# An optochemical tool for light-induced dissociation of adherens junctions to control mechanical coupling between cells

Dirk Ollech [1,2,6], Tim Pflästerer[1,2,3], Adam Shellard[4], Chiara Zambarda [1,2], Joachim Pius Spatz [1,2], Philippe Marcq [5], Roberto Mayor[4], Richard Wombacher[3,7]* & Elisabetta Ada Cavalcanti-Adam [1,2,7]*

The cadherin-catenin complex at adherens junctions (AJs) is essential for the formation of cell-cell adhesion and epithelium integrity; however, studying the dynamic regulation of AJs at high spatio-temporal resolution remains challenging. Here we present an optochemical tool which allows reconstitution of AJs by chemical dimerization of the force bearing structures and their precise light-induced dissociation. For the dimerization, we reconstitute acto-myosin connection of a tailless E-cadherin by two ways: direct recruitment of α-catenin, and linking its cytosolic tail to the transmembrane domain. Our approach enables a specific ON-OFF switch for mechanical coupling between cells that can be controlled spatially on subcellular or tissue scale via photocleavage. The combination with cell migration analysis and traction force microscopy shows a wide-range of applicability and confirms the mechanical contribution of the reconstituted AJs. Remarkably, in vivo our tool is able to control structural and functional integrity of the epidermal layer in developing *Xenopus* embryos.

[1] Department of Cellular Biophysics, Max Planck Institute for Medical Research, Jahnstraße 29, D-69120 Heidelberg, Germany. [2] Department of Biophysical Chemistry, Institute of Physical Chemistry, Heidelberg University, INF 253, D-69120 Heidelberg, Germany. [3] Department of Pharmaceutical Chemistry, Institute of Pharmacy and Molecular Biotechnology, Heidelberg University, INF 364, D-69120 Heidelberg, Germany. [4] Department of Cell and Developmental Biology, University College London, London WC1E 6BTUK. [5] PMMH, CNRS, ESPCI Paris, PSL University, Sorbonne Université, Université de Paris, F-75005 Paris, France. [6]Present address: Applied Physics Department, Science for Life Laboratory and KTH Royal Technical University, Tomtebodavägen 23A, S-17165 Stockholm, Sweden. [8]These authors contributed equally: Richard Wombacher, Elisabetta Ada Cavalcanti-Adam. *email: wombacher@uni-heidelberg.de; eacavalcanti@mr.mpg.de

Adherens junctions (AJs) are important structures for the maintenance of cell–cell adhesion in the epithelium. They are dynamically regulated in fundamental physiological and pathological processes like embryonic development, wound healing and cancer metastasis[1]. The transmembrane protein E-cadherin forms homophilic contacts with E-cadherins of adjacent cells via the extracellular domain, whereas the cytosolic tail domain is connected to the contractile acto-myosin network via various members of the catenin protein family[2,3]. The minimal endogenous AJ complex consists of β-catenin linking E-cadherin to α-catenin which is binding F-actin. However, direct fusion of actin binding sites (ABS) to tailless E-cadherin can reconstitute AJs in knock-out cells[4–8]. Dynamic regulation of AJ formation and dissociation is essential for homeostasis and functional integrity of epithelial tissues. These mechanisms have been studied using constitutive or inducible knockout strategies[9], or by preventing E-cadherin interaction either using antibodies[10,11] or Ca$^{2+}$ depletion[11,12]. Although these tools gave remarkable insights into AJ formation dynamics, they are limited in temporal resolution and lack spatial control.

Here, we present LInDA (Light-Induced Dissociation of Adherens Junctions), an optochemical tool based on photocleavable dimerizers[13–16] for the precise control of E-cadherin-mediated AJs dynamics (Fig. 1a). In LInDA we use small light-controlled molecules (Fig. 1b) to make benefit of the modular architecture of AJs and steer the cadherin-catenin complex formation (Fig. 1c). For this, we developed two setups to control with LInDA the assembly and disassembly of AJs in epithelial cells which are lacking these structures (either from knockout or due to transformation-associated loss in protein expression). (i) To establish the E-cadherin/α-catenin linkage we replaced the cytosolic tail domain of E-cadherin with Halo tag[17] (E-cadherin-Δcyto-Halo) and the N-terminal domain of α-catenin with SNAP tag[18] (SNAP-ΔN-α-catenin) (Fig. 1d and Supplementary Fig. 1a–d). (ii) To reconstitute the E-cadherin protein, we combined the expression of E-cadherin-Δcyto-Halo with the expression of the cytosolic tail of E-cadherin presenting a DHFR tag[19] (DHFR-cyto) (Fig. 1e and Supplementary Fig. b, e–g).

## Results

**LInDA facilitates an ON-OFF switch for adherens junctions.** The two constructs shown in Fig. 1d were coexpressed in human epidermoid carcinoma cells (A431 cells) knocked out for α-catenin. This allows the chemically-induced reconstitution of AJs by addition of the photocleavable small molecule Ha-pl-BG or photostable Ha-BG (Fig. 2a–c and Supplementary Fig. 2). Since both E-cadherin-Δcyto-Halo and SNAP-ΔN-α-catenin are lacking the β-catenin binding sites, the two proteins cannot interact with each other without the cell-permeable chemical dimerizers. Therefore, AJs are not assembled, as evidenced by the absence of E-cadherin, α-catenin and β-catenin clustered colocalization, and by lack in association of actin filaments at cell–cell contacts (Fig. 2a left column). Upon dimerizer binding, the ABS carrying α-catenin construct is recruited to the tailless E-cadherin at the cell membrane, forming stable E-cadherin-α-catenin complexes (Fig. 2a right column, first and second panel from top; Fig. 2c lane 2). The complexes accumulate at the cell–cell interfaces, stabilizing endogenous E-cadherin by lateral clustering, which in turn triggers the recruitment of additional AJ components, like e.g., β-catenin to the membrane and inducing the reorganization

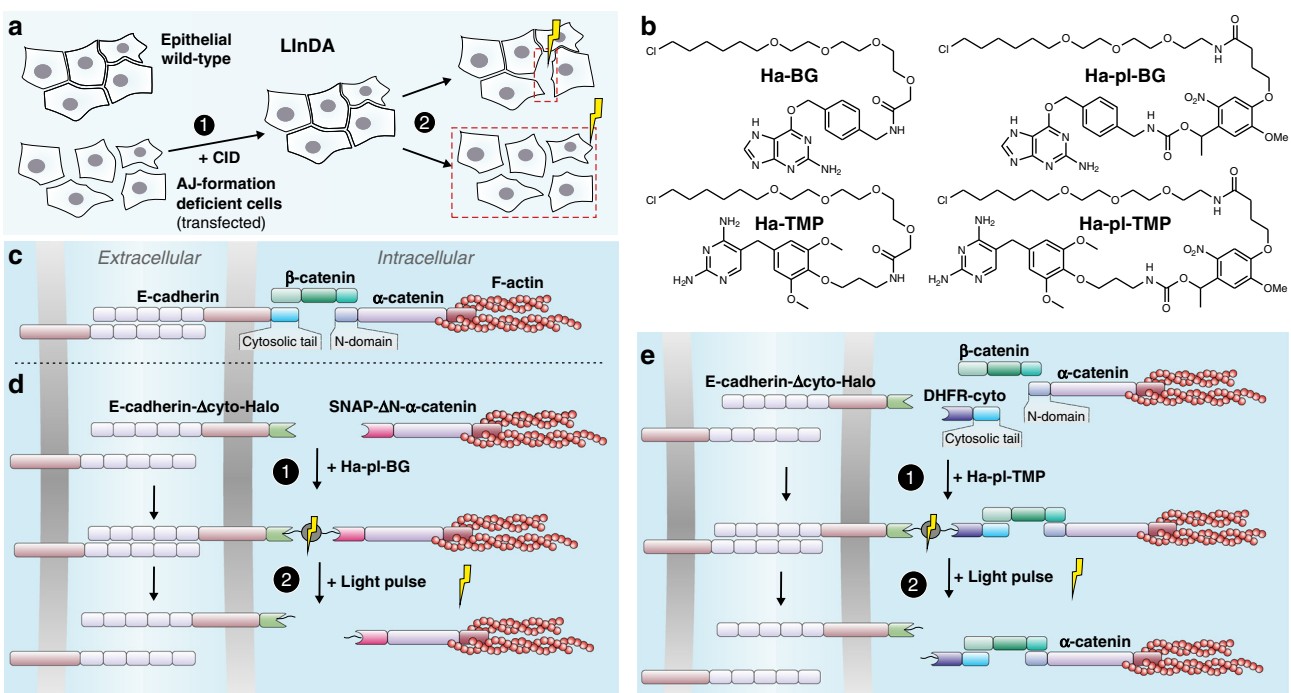

**Fig. 1 Design of the setup for Light-Induced Dissociation of Adherens junctions (LInDA). a** The chemically induced dimerization (CID, 1) of two different proteins or of two parts of a split variant of a protein at adherens junctions (AJ) leads to formation of functional AJs and cell compaction. The dimeric complex is cleaved with light of appropriate wavelength (350–405 nm) in subcellular regions or larger areas (2), leading to AJ dissociation and cell dissemination. **b** Chemical structures of the photostable dimerizer Ha-BG and Ha-TMP, and the photocleavable dimerizer Ha-pl-BG and Ha-pl-TMP. **c** In endogenous AJ, β-catenin facilitates the connection between E-cadherin and α-catenin. **d** The β-catenin binding domains of E-cadherin and α-catenin were replaced by Halo tag and SNAP tag, respectively. Addition of Ha-pl-BG induces the formation of an E-cadherin-α-catenin heterodimer; application of 350–405 nm light pulse leads to complex dissociation. **e** The E-cadherin receptor was split in two parts, where the cytosolic juxtamembrane domain is fused with a Halo tag and the cytosolic tail fused to a DHFR tag. Addition of Ha-pl-TMP induces the reconstitution of the receptor, the binding of β-catenin at AJs and the further recruitment of α-catenin/actin complex. The dissociation is then triggered by application of 350–405 nm light pulse.

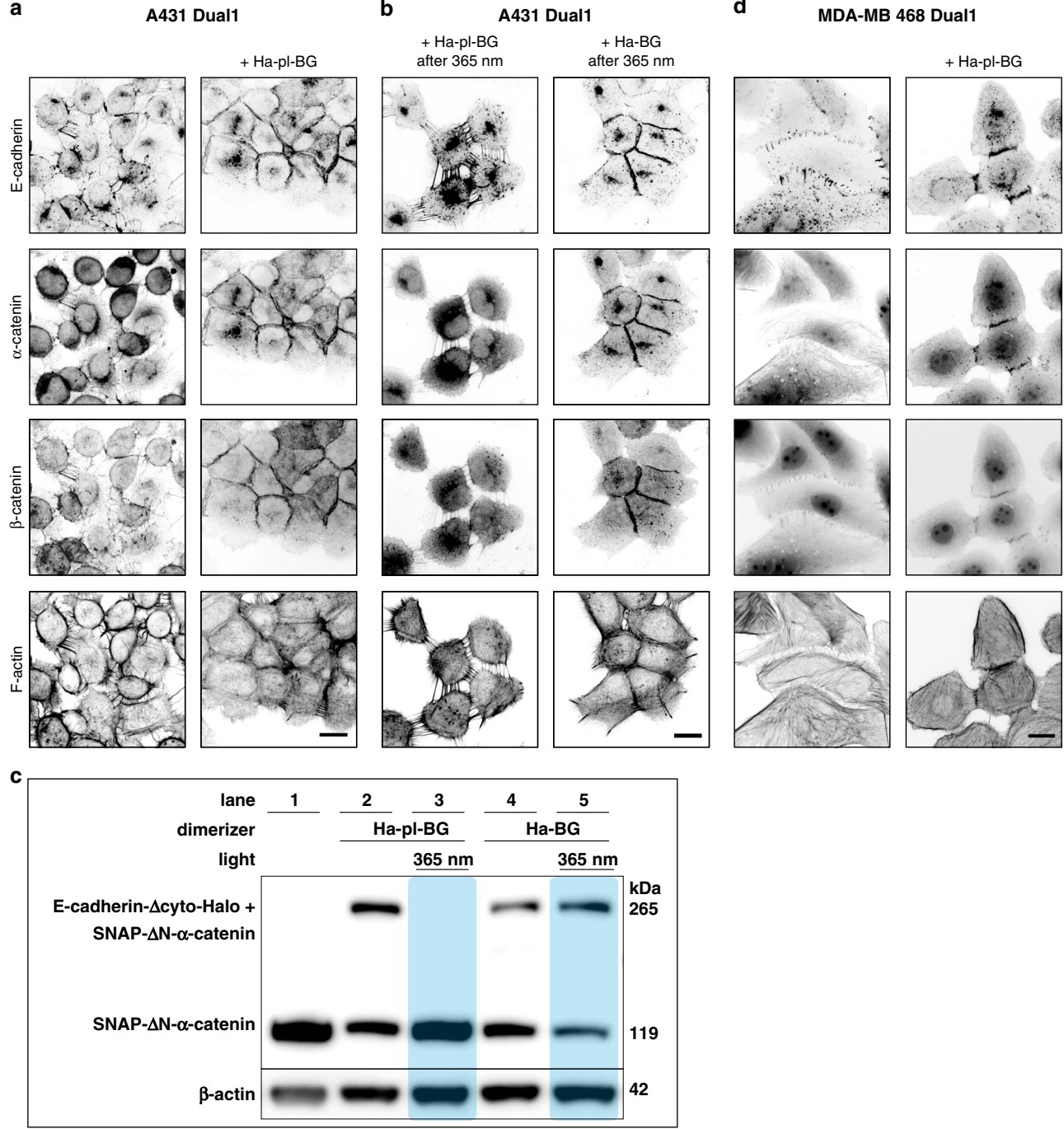

**Fig. 2 Establishing a functional link between E-cadherin and α-catenin with optochemical dimerizers. a, b** Immunofluorescence images of A431 α-catenin KO cells coexpressing E-cadherin-Δcyto-GFP-Halo and SNAP-mCherry-ΔN-α-catenin. Without dimerizer E-cadherin shows diffuse membrane localization, whereas α-catenin and β-catenin are cytosolic. Dimerizer induced E-cadherin-α-catenin complex formation indirectly recruits also β-catenin and causes rearrangements of actin fibers. 365 nm light disseminates Ha-pl-BG treated cells but not Ha-BG treated cells. Scale bars 20 μm. **c** Western Blot analysis of E-cadherin-α-catenin complexes. Without dimerizer (lane 1) only the 119 kDa SNAP-(mCherry)-α-catenin is detected, whereas with dimerizer (lane 2 and 4) an additional band of 265 kDa is detected, resembling the hetero-dimer E-cadherin-α-catenin complex. After 365 nm light, the heavy band disappears in Ha-pl-BG treated cells but not in Ha-BG samples (lane 3 and 5). β-actin serves as loading control. **d** Application of LInDA to dissociate AJs in MDA-MB-468 epithelial cancer cells coexpressing E-cadherin-Δcyto-GFP-Halo and SNAP-mCherry-ΔN-α-catenin. Filamentous actin was labeled using phalloidin conjugated with Alexa647 dye. In absence of the dimerizer, punctate E-cadherin clusters are present at the cell membrane, whereas α-catenin and β-catenin are diffuse in the cytoplasm. Addition of Ha-pl-BG induces E-cadherin mediated AJ formation by recruiting the cytosolic α-catenin and β-catenin to the cell membrane. Scale bar 20 μm.

of actin fibers (Fig. 2a second column, third and fourth panel from top). Illumination with near UV light cleaves the Ha-pl-BG dimerizer and leads to immediate release of the α-catenin construct from the AJs, accompanied by loss of cell–cell contacts (Fig. 2b left column, Fig. 2c lane 3). AJs that have been reconstituted using the non-cleavable Ha-BG dimerizer are not affected by UV illumination (Fig. 2b right column; Fig. 2c lane 4 and 5) clearly indicating that LInDA mediated AJ dissociation is a specific effect resulting from Ha-pl-BG cleavage.

We tested the applicability of LInDA to induce AJs formation also in epithelial cancer cells (MDA-MB-468 derived from human breast carcinoma) which do not express α-catenin but still present normal endogenous expression of E-cadherin and β-catenin. It has been previously shown[5,20] that reintroduction of α-catenin in these cells leads to reinforcement of E-cadherin-mediated junctions and anchorage to the actin cytoskeleton. The localization of the GFP-tagged E-cadherin-Δcyto-Halo at the cell membrane takes place in both cases, i.e., without and with the Ha-pl-BG dimerizer (Fig. 2d). However, the type of junctions that are formed differ in terms of morphology and distribution of E-cadherin along the junction, being the protein rather localized in punctate and radial clusters in absence of the dimerizer (Fig. 2d left column). In presence of the dimerizer E-cadherins are arranged in linear defined junctions between adjacent cells (Fig. 2d right column). As observed for the A431 α-catenin deficient cells, mCherry-tagged SNAP-ΔN-α-catenin and immunostained β-catenin localize at AJs only in presence of the dimerizer (Fig. 2d right column, second and third panel from top); this leads also to reorganization of the actin cytoskeleton with parallel robust fibers at AJs sites in the cells treated with the dimerizer (Fig. 2d right column, panel from top). With the dimerizers it is therefore possible to assemble stable cell–cell contacts mediated by E-cadherin and that this is due to the specific recruitment of α-catenin and β-catenin, and subsequent coupling to the actin cytoskeleton.

**Subcellular and tissue scale manipulation of AJs.** We next investigated the flexibility of LInDA to address AJs in epithelial cell cultures at different scales in time and space. The E-cadherin-Δcyto-Halo and SNAP-ΔN-α-catenin coexpressing A431 α-catenin KO cells can be followed for several hours without observing cell death or reassembly of AJ. For dissociation of AJs in subcellular regions (Fig. 3a and Supplementary Movie 1), cells were pre-incubated with Ha-pl-BG overnight to form defined linear AJs (Fig. 3a upper row, indicated as time point −01:00 min). To achieve high spatial control for the irradiation and photocleavage we used a laser scanner module equipped with a 405 nm laser and targeted AJs at the cell–cell interface of only two adjacent cells (area in white dashed frame, Fig. 3a). While AJ dissociation leads to detachment of the cells from each other in the irradiated area, E-cadherin-α-catenin colocalization and cell–cell attachment at the non-irradiated AJs is not affected (05:00 min). Interestingly, E-cadherin localization remains fairly stable until the cells start to detach from each other (approximately 30 s after irradiation), as shown in the kymograph analysis (Fig. 3b and Supplementary Movie 1). This suggests that E-cadherins which lost the connection to the actomyosin network, tend to stay clustered, but cannot exert sufficient adhesive interactions to keep the cells attached to each other.

We also determined the effect of LInDA on epithelial monolayer formation (Fig. 3c and Supplementary Movie 2). The cells were seeded at high density in a glass bottom dish. When imaged in phase contrast mode, cells show a strong halo due to high diffraction at the cell–cell interfaces. Notably, at full confluency, these cells do not stop to proliferate. However,

dividing cells cannot reintegrate into the adherent monolayer after undergoing cytokinesis and stay loosely attached on top of the adherent cells, showing a spherical shape. Upon addition of Ha-pl-BG (Fig. 3c, 00:00 h), cells start to form AJs and the monolayer undergoes compaction, characterized by reduced scattering of light around the cell body in phase contrast images (Fig. 3c, 04:00 h). Cells that were loosely attached after cytokinesis due to lack of space, reintegrate into the adherent monolayer. We then used the 405 nm laser to cleave the dimerizer and consequently triggered AJs disassembly in a defined area (dashed line framed area in Fig. 3c, 04:05 h). The cells within the targeted area immediately disseminate and push out the excess of cells that have been integrated during the compaction, whereas the cells in the untreated area retain their morphology and are integrated in a compact epithelial monolayer. Therefore, LInDA is a powerful tool to study epithelial monolayer compaction and dynamics, e.g., during migration, unjamming or cell extrusion processes.

**Molecular assembly and disassembly of AJ with LInDA.** LInDA facilitates two subsequent switches that allow the targeted study of AJ dynamics in cellular assemblies: The ON switch by chemically induced dimerization and the light-induced OFF switch, which allows ultra-high temporal and spatial resolution. Remarkably, when we apply LInDA to α-catenin knockout cells, we do not observe any basal AJ formation in the absence of the small molecule dimerizer or after light irradiation, indicating that LInDA represents a binary switch to study the effects of AJ formation and dissociation. The assembly of AJs induced by LInDA was monitored by time-lapse video microscopy of cells treated with the Ha-pl-BG or Ha-BG (Fig. 4a, Supplementary Fig. 3a, and Supplementary Movie 3, 4) to establish the link between E-cadherin and α-catenin. The fluorescence intensity of GFP-E-cadherin and mCherry-α-catenin was monitored over time and the normalized intensity values of the GFP and mCherry signals along the cross-section of the contact area drawn between two adjacent cells (white dashed line in Fig. 4a, and Supplementary Fig. 3a) were plotted (Fig. 4b, and Supplementary Fig. 3b). The assembly of AJs induced by the Ha-pl-BG and Ha-BG dimerizer is characterized by the initial recruitment of α-catenin to the cell membrane and accumulation of the E-cadherin-α-catenin complexes in clusters followed by the appearance of continuous structures between two adjacent cells. Over time, the E-cadherin mediated structures mature into straight AJs, characterized by colocalization of the two proteins. Following the treatment with 350 nm light, cells incubated with Ha-pl-BG (Fig. 4a and Supplementary Movie 3) immediately impair cell–cell contacts, whereas in presence of the Ha-BG dimerizer cell–cell contacts are maintained and no major changes in E-cadherin and α-catenin localization are observed (Supplementary Fig. 3 and Supplementary Movie 4), further supporting the observation that the disassembly of AJs is specifically mediated by light-induced cleavage of the dimerizer. During disassembly due to photocleavage of Ha-pl-BG dimerizer, the cell bodies and cell–cell contact undergo major reorganization, resulting in displacement of cells, loss of AJ linearity and appearance of tethers between adjacent cells. At the molecular level, α-catenin is rapidly displaced from AJs and it localizes in the cytosol, whereas the GFP signal of the E-cadherin is still found at the cell membrane but mainly localized in the tethers (Fig. 4a, b, and Supplementary Movie 3).

To investigate the nature of the tethers, we stained the A431 α-catenin knockout cells coexpressing E-cadherin-Δcyto-Halo and SNAP-ΔN-α-catenin cells for components of desmosomes, another type of anchoring junction that connects the intermediate filament network of neighboring cells (Fig. 4c). In absence of the

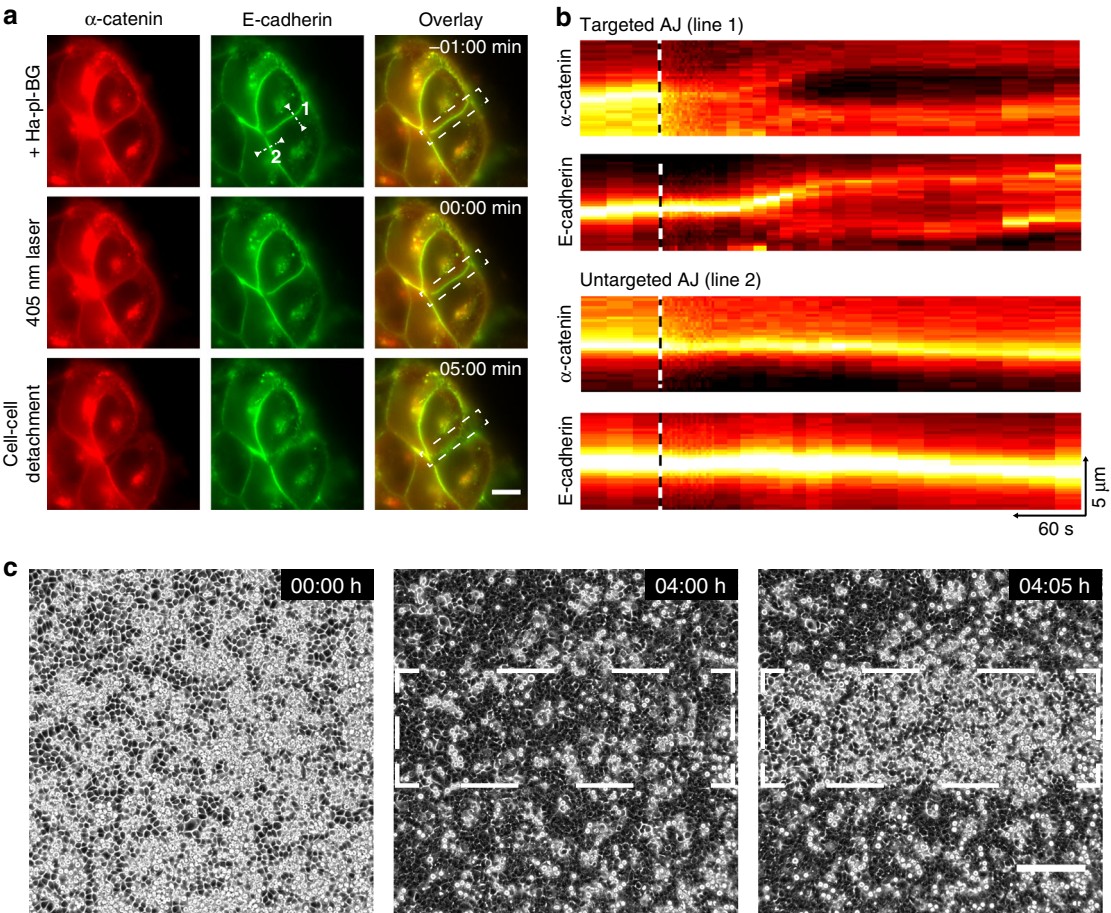

**Fig. 3 Multiscale targeting of AJs assembly and disassembly with high spatial precision. a** For visualization in live cell experiments, E-cadherin (shown in green) and α-catenin (shown in red) are tagged with GFP and mCherry, respectively. Targeted dissociation of Ha-pl-BG reconstituted AJs with subcellular precision before (upper row) directly after (middle) and 5 min after 405 nm laser area scanning (white dashed rectangular in the overlay). Scale bar 10 μm. **b** Kymograph analysis of E-cadherin and α-catenin intensities of targeted (line 1) and untargeted (line 2) AJs in **a**. The time point of laser scanning is indicated as black and white dashed lines. Scale as indicated by arrows in the lower right. **c** Monolayer compaction via AJ reconstitution after addition of Ha-pl-BG leads to reintegration of the loosely attached cells in a monolayer. After 4 h, the white dashed outlined area was scanned with a 405 nm laser. Note that only cells in the targeted area change their morphology and push out excess cells immediately. Scale bar 200 μm.

dimerizer (Fig. 4c first row), the intermediate filament protein cytokeratin is loosely associated with E-cadherin aggregates that form tethers between adjacent cells, but do not accumulate desmoplakin i.e., do not form stable desmosomal contacts. In contrast, induction of AJ assembly by addition of the dimerizer leads to formation of defined desmosomal plaques characterized by the accumulation of desmoplakin clusters between cytokeratin fibers of neighboring cells (Fig. 4c second row). Following the light-induced dissociation of AJs and retraction of cells, the tethers become more evident and still retain cytokeratin and desmoplakin clusters (Fig. 4c third row). This confirms that the immediate effect of LInDA caused by administration and photocleavage of the dimerizer is acting solely on AJs, while only the presence of AJs allows the formation of desmosomes. Taken together these results indicate that the specificity of LInDA in modulating the assembly and disassembly of AJs allows monitoring of molecular dynamics which are involved in the functional stability and maturation of cell–cell contacts.

**LInDA modulates collectivity in epithelial cells**. To test if AJ formation and epithelial layer compaction induced by LInDA would reflect on collective dynamics, we cultured the cells with or without addition of dimerizer in a confinement to form a confluent monolayer. Upon removal of the confinement, cells start to migrate into the free area (Fig. 5a and Supplementary Movie 5 and 6). To analyze the collectivity of this migration, we measured the velocity field between two time points by particle image velocimetry (PIV) analysis and calculated the correlation length of the lateral velocity component (i. e. along the migration front), as previously described[21] (Supplementary Fig. 4). We found a significant difference ($p < 0.0001$, by unpaired $t$-test with Welch´s correction) between the lateral velocity correlation lengths being 258.0 ± 4.9 μm (mean ± s.d.) for cells treated with dimerizer and 165.5 ± 1.4 μm for cells migrating in absence of dimerizer. This indicates stronger collectivity and coordination in the migratory behavior of cells treated with Ha-pl-BG.

Next we determined whether the higher collectivity is specific to cells with reconstituted AJs and can be reduced by cleaving the dimerizer. Therefore, we recorded the migration of a cell layer preincubated with Ha-pl-BG for 2 h before cleaving the dimerizer in a defined region at the migration front and continued the time lapse recording (Fig. 5b and Supplementary Movie 7). The average velocity correlation length in the targeted region is 184.1 ± 5.5 μm before and 136.1 ± 2.7 μm after 405 nm laser scanning, whereas the velocity correlation length in untargeted regions remains unvaried (Supplementary Fig. 5). It should be noted that although the initial average velocity correlation length

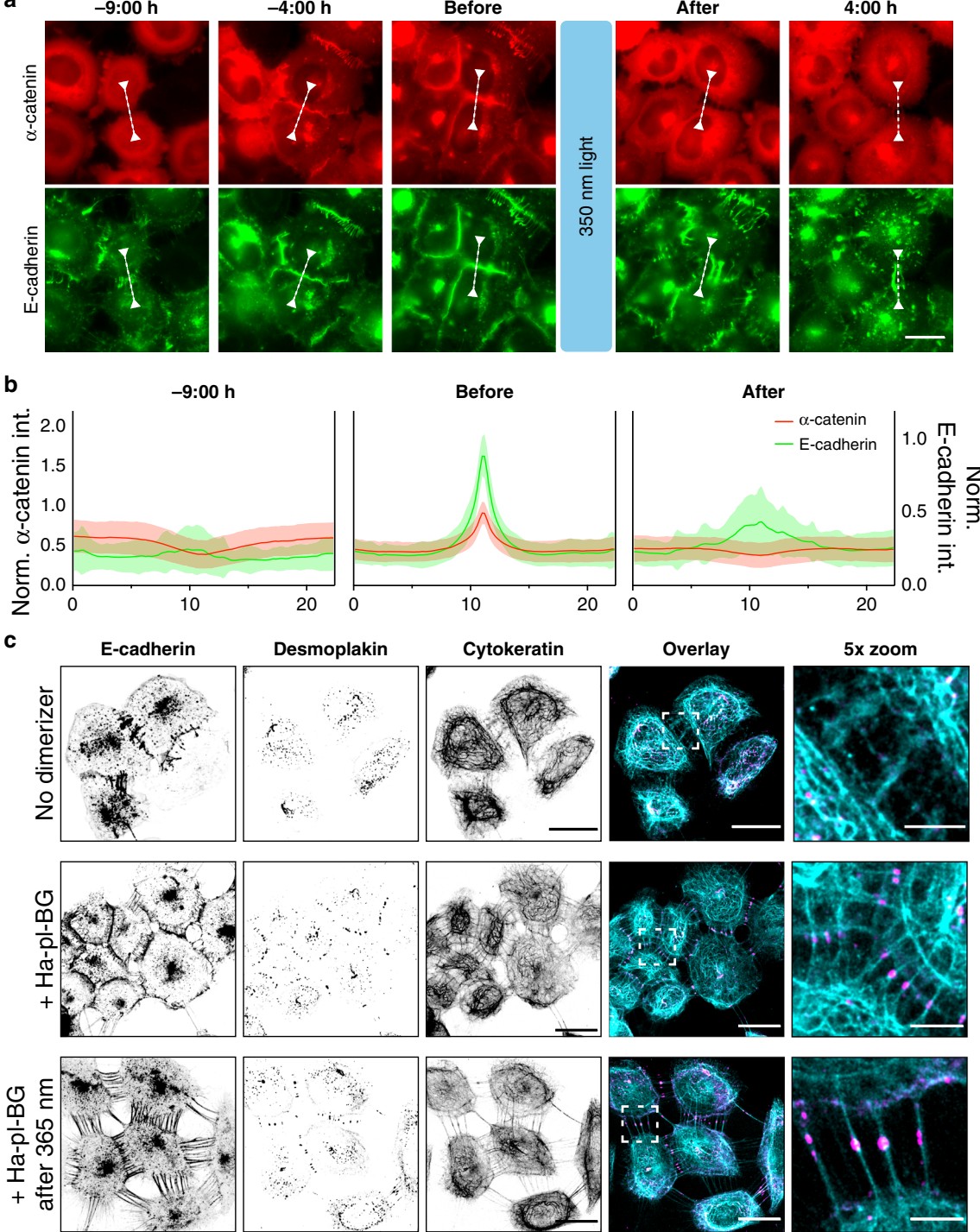

**Fig. 4 Dimerizer-mediated reconstitution followed by LInDA to specifically target AJs. a** When coexpressed in α-catenin KO cells, the α-catenin (mCherry-labled, shown in red) construct is cytosolic, whereas E-cadherin (GFP-labled, shown in green) is located at the cell membrane forming unstable complexes at cell–cell-interfaces. Following addition of Ha-pl-BG (−9:00 h) AJs form (−4:00 h) and mature into defined linear structures accompanied by cell compaction (before). After a short pulse of 350 nm light the E-cadherin-α-catenin complex disassembles and α-catenin becomes cytosolic again (after and 4:00 h). E-cadherin clusters destabilize and cells disseminate. Scale bar 20 μm. **b** Normalized profiles for GFP (green) and mCherry (red) fluorescence intensity in cross-sections of cell–cell contacts as measures for localization of E-cadherin and α-catenin, respectively. A representative cross-section is shown by the arrow headed dashed lines in **a**. The lines have been repositioned for each time point to reflect profiles perpendicular and centered to the cell–cell interface. Mean ± s.d. are shown for *n* = 45 cross-sections in multiple fields of view for each time point. **c** Staining for desmosome proteins reveals the presence of desmoplakin at cell–cell contacts only in presence of the dimerizer and in cell tethers following the light-induced dissociation of AJs. Scale bar 20 μm for single color images and overlay image, 5 μm for 5× zoom image.

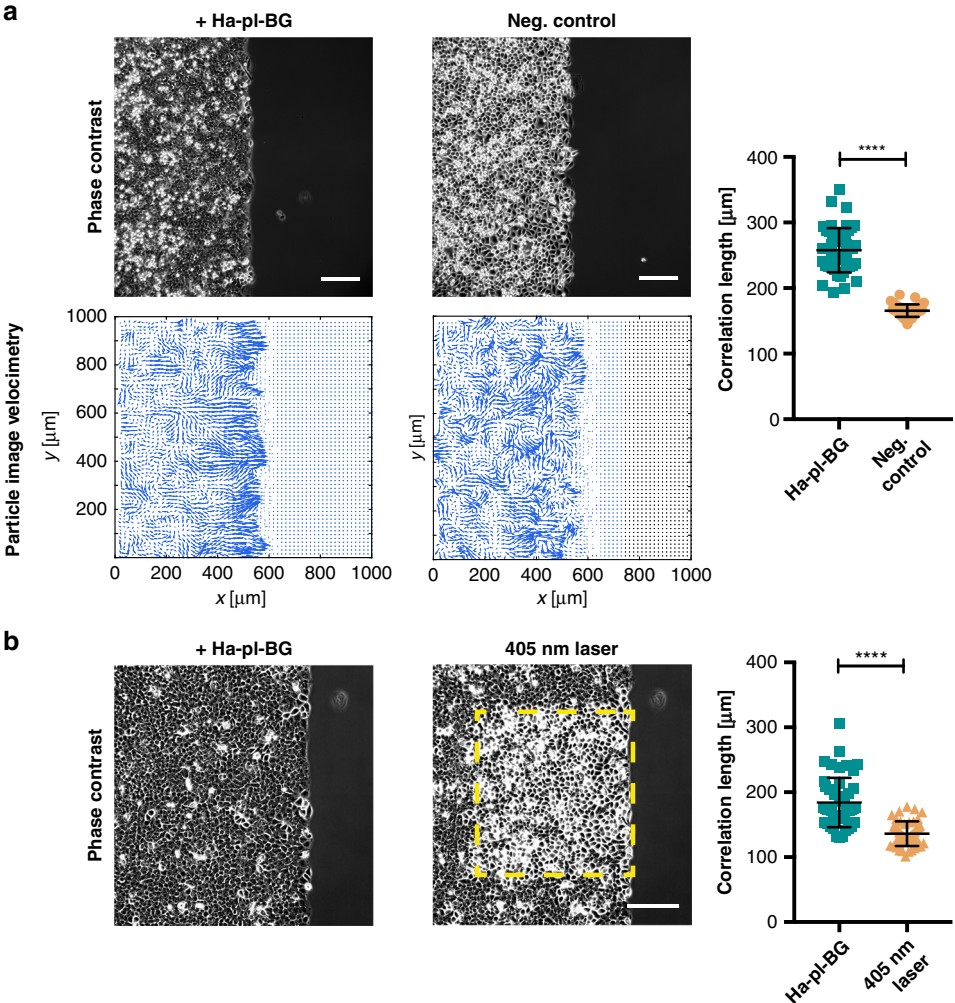

**Fig. 5 LInDA has an impact on the collective behavior of epithelial monolayers. a** PIV analysis of cell migration. Vector fields of cell velocities from two consecutive frames were calculated from phase contrast images to determine the lateral velocity correlation length as a measure of collectivity. Cells treated with Ha-pl-BG show a significantly higher velocity correlation length then the untreated control cells (neg. control). Scale bar 200 μm. $n = 192$ (48 time points from 4 positions) for cells with dimerizer (dark green squares) and untreated control (orange circles), respectively. ****$p < 0.0001$ by unpaired $t$-test with Welch´s correction. **b** PIV analysis of dimerizer treated cells before and after light-induced dissociation of AJs shows a significant reduction of collectivity in the targeted cells as measured by the velocity correlation length. Scale bar 200 μm. $n = 48$ (12 time points from 4 positions) for cells before (dark green squares) and after 405 nm exposure (orange circles), respectively. ****$p < 0.0001$ by unpaired $t$-test with Welch's correction.

in the migrating epithelial layer is affected by cell density, the drastic change induced by the photocleavage of the dimerizer leads in any case to a significant reduction of the average velocity correlation length.

We combined LInDA with traction force microscopy (TFM) analysis to determine the amount of stresses transmitted across the adherent monolayer upon dimerizer induced AJ assembly. Cells were seeded in a confinement on a polyacrylamide gel (Young's Modulus 200 Pa) containing fluorescent beads and functionalized with covalently bound collagen type I to facilitate cell adhesion and incubated with Ha-pl-BG to form a tightly packed epithelial layer (Fig. 6) or without the Ha-pl-BG dimerizer (Supplementary Fig. 6). After release of the cells from the confinement, we followed cell migration and gel deformation by analysis of bead displacement in $xy$ to determine the traction forces before and after light-induced cleavage of the dimerizer (Fig. 6a, b and Supplementary Movie 8). Cells with reconstituted AJs showed a defined migration front with tractions that progressively increase during the first time points and seem to reach steady state after 45–60 min (Supplementary Fig. 7a).

Following the cleavage of the dimerizer, after 2 h a large number of cells was extruded from the migrating layer. Although the morphology of the cell layer changes immediately after AJ dissociation and the straight migration front rapidly disappears, tractions at the migration front decrease only gradually and become irregularly distributed across the cell layer. The average traction force normalized is $2.1 \pm 0.4$ Pa before (positions analyzed $n = 5$) and becomes $1.7 \pm 0.4$ Pa ($n = 3$) after photocleavage (Supplementary Fig. 7a). In absence of the dimerizer, tractions slowly decrease over time with an average value of the traction force norm $1.4 \pm 0.2$ Pa ($n = 3$) (Supplementary Fig. 7b).

The advancing monolayer is constantly evolving and its mechanical behavior may be also affected by active pushing out the extruded cells from the adherent layer, loss of mechanotransducing AJs or a combination of both. By applying Bayesian inversion stress microscopy[22], we therefore estimated the internal stress field of the epithelial cell layer after the dimerizer-induced formation of AJs and following the light-induced dissociation of AJs (Fig. 6a, c). The tension in the monolayer was determined by averaging the monolayer in bulk over the period of

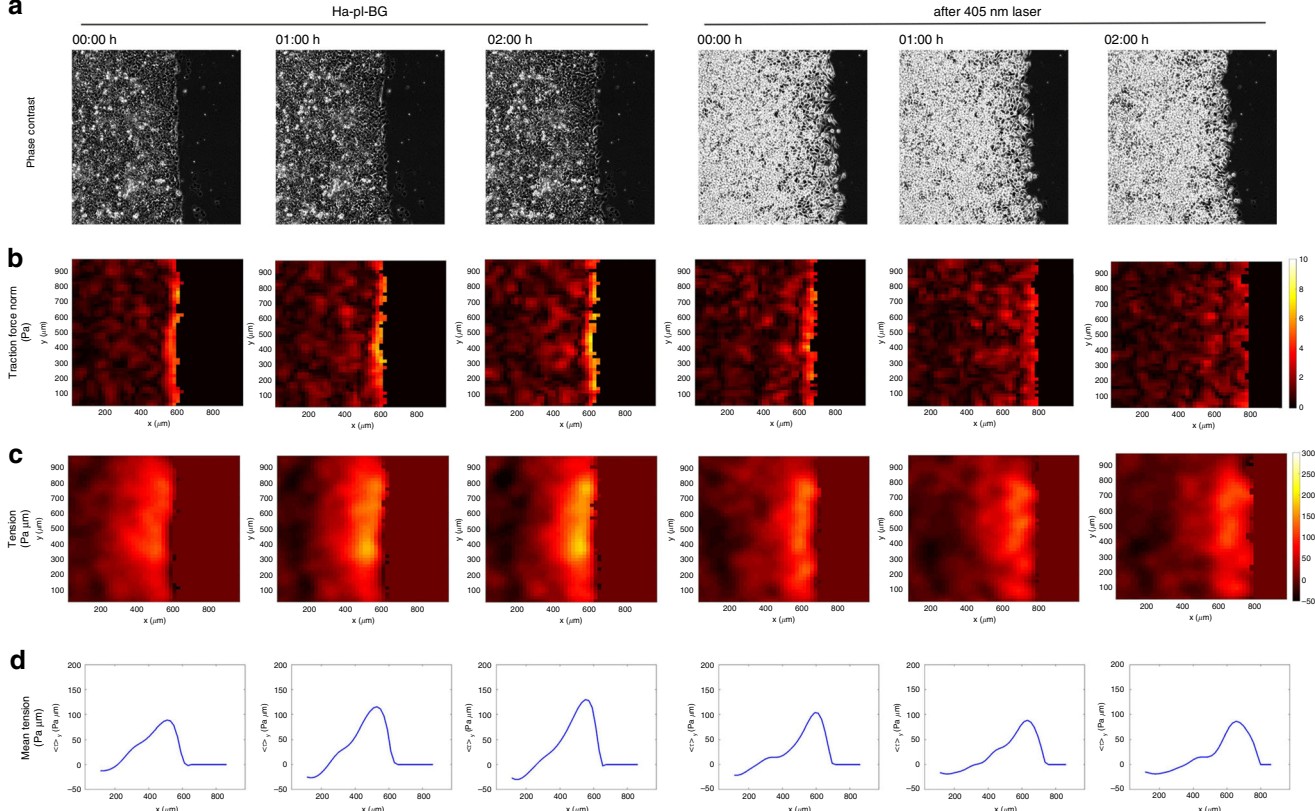

**Fig. 6 LInDA changes the internal tension of epithelial monolayers. a** Phase contrast images of migrating cells treated with Ha-pl-BG. Shown are three selected time points from a continuous time lapse recording before (Ha-pl-BG) and after cleaving the dimerizer (after 405 nm laser). **b** Heat maps of traction force norm in the same spatial domain as **a**. Traction Force Microscopy analysis of migrating cells shows that the norms of traction forces are gradually reduced after photocleavage of the Ha-pl-BG dimerizer. **c**, **d** Internal stresses are estimated applying Bayesian Inversion Stress Microscopy (BISM) on traction force data. BISM analysis shows that the monolayer tension is also reduced after exposure of the migrating monolayer to UV light pulse. **c** Heat maps of the mechanical tension in the same spatial domain as in **a**. **d** 1D profile of the tension averaged over the direction parallel to the moving front.

observation. Its estimate is $50.8 \pm 6.4$ Pa μm (field of view $n = 5$) for 2D migrating dimerized epithelial layer and it drops to $27.5 \pm 6.7$ Pa μm ($n = 3$) after photocleavage of the dimerizer and subsequent disassembly of AJs. The 1D profiles of the averaged monolayer tension parallel to the moving front also indicate the increase in mean tension at the front after dimerizer induced AJ assembly as the monolayer starts to migrate (Fig. 6d). This is followed by a gradual drop in mean tension following photocleavage of the dimerizer (Supplementary Fig. 7a). If AJs are not reconstituted by dimerizer treatment (Supplementary Fig. 6), the tension in the cell monolayer and slowly decreases over time, with an average value of $8.1 \pm 5.5$ Pa μm ($n = 3$) (Supplementary Fig. 7b).

**LInDA disassembles E-cadherin-mediated contacts in vivo.** To minimize the manipulation of endogenous proteins and exploit the in vivo application of our dimerizers, we reconstituted an E-cadherin split protein and guided the assembly and disassembly of AJs induced by LInDA. The reconstitution of a functional E-cadherin induced by the optochemical dimerizer Ha-pl-TMP was first validated in MDCK cells knocked out for the endogenous expression of E-cadherin. Cells expressing the E-cadherin-Δcyto-Halo and the DHFR-tagged and GFP-tagged cytosolic tail of E-cadherin (indicated as DHFR-cyto) were stained for α-catenin and F-actin (Supplementary Fig. 8 first row). Since cell junctions are maintained by other cadherins expressed in these cells[23], α-catenin and actin are still evident at cell–cell contacts, although E-

cadherins are not dimerized. However, in presence of Ha-pl-TMP dimerizer, the GFP signal of DHFR-cyto translocates to the cell membrane and colocalizes with α-catenin; further, actin filaments become more evident (Supplementary Fig. 8 second row). Following photocleavage of the dimerizer, the cytosolic tail of E-cadherin accumulates in the cytoplasm, whereas α-catenin and actin are still present at cell junctions and cells do not completely dissociate from each other (Supplementary Fig. 8 third row). This shows that LInDA acts specifically on the targeted E-cadherin without disrupting the functionality of other cadherin-catenin complexes.

To test the in vivo relevance of LInDA, mRNA encoding E-cadherin-Δcyto-Halo and DHFR-cyto were co-injected into *Xenopus* embryos, which were then grown and incubated with dimerizer and subjected to 405 nm exposure (Fig. 7a). Unlike uninjected controls (Fig. 7b, c), epithelial dissociation was observed from developmental stage 10.5 in embryos expressing E-cadherin-Δcyto-Halo and DHFR-cyto (Fig. 7d, e). At this time, endogenous E-cadherin begins to be expressed[24], suggesting that the constructs act as a dominant negative. Dissociation was rescued by incubation of the embryos with the dimerizer Ha-pl-TMP (Fig. 7f, g). Photocleavage with a 405 nm laser induced dissociation of the epithelial layer (Fig. 7h, i). Using confocal microscopy, we validated the efficient translocation of DHFR-cyto to the cell contact when embryos were incubated with Ha-pl-TMP (Fig. 7m). Whereas a 405 nm laser caused cytoplasmic relocation of DHFR-cyto when embryos were incubated with

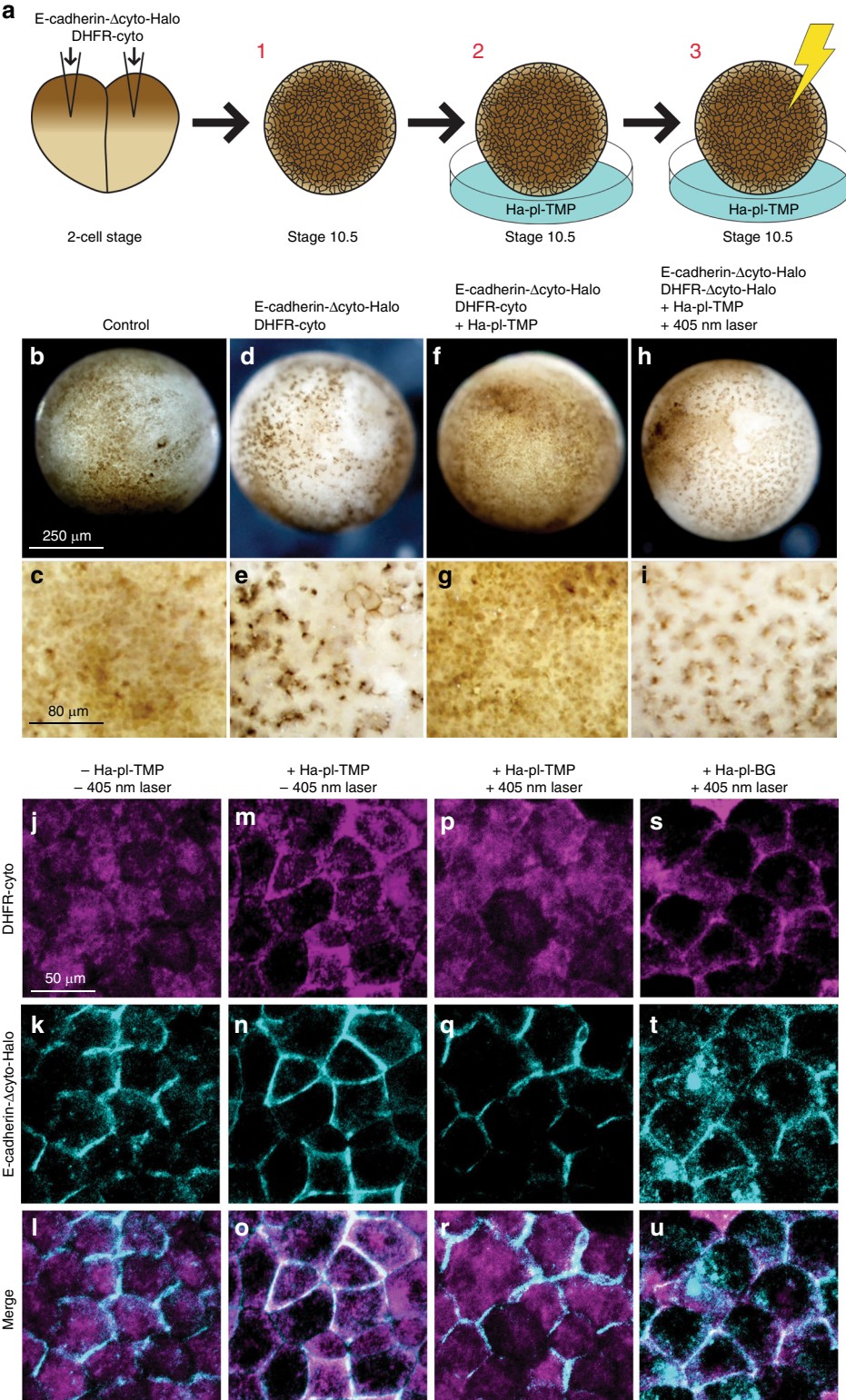

**Fig. 7 LInDA tools affect epithelial integrity in vivo. a** Diagram of the experimental setup. Xenopus embryos were injected with DHFR-cyto and E-cadherin-Δcyto-Halo at the two-cell stage and imaged at stage 10.5, with or without the dimerizer and with or without exposure to a 405 nm laser. **b–i** Epidermal dissociation was observed by co-injection of DHFR-cyto and E-cadherin-Δcyto-Halo (**d, e**) compared to wild type controls (**b, c**), which was rescued by incubation with the dimerizer (**f, g**). Dissociation was induced by photocleavage under a 405 nm laser (**h, i**). **j–u**, DHFR-cyto is cytosolic and E-cadherin-Δcyto-Halo is localized to the cell contact (**j–l**). Upon the addition of Ha-pl-TMP dimerizer, DFHR-cyto translocates to the cell contact (**m–o**), which can be disrupted by exposure to blue light (**p–r**). Blue light fails to prevent accumulation of DHFR-cyto at the cell contact when embryos are incubated with the non-photocleavable Ha-TMP dimerizer (**s–u**).

Ha-pl-TMP (Fig. 7p), it did not have this effect when embryos were incubated with the non-photocleavable Ha-TMP (Fig. 7s). These results demonstrate that LInDA is an effective and relevant manipulator of E-cadherin in vivo.

## Discussion

LInDA is offering two controllable, binary switches to study AJs: the chemically induced reconstitution of AJ via addition of the dimerizer as a systemic ON switch and the spatio-temporally precise OFF switch with short pulses of 405 nm or near UV light. Thus, it allows the selective modulation of single AJs to study the molecular dynamics and temporal recruitment of AJ proteins during assembly and disassembly of cell–cell contacts, and further the changes in cell-matrix and cell–cell forces. LInDA is far superior to methods applied previously to study AJ, e.g., methods based on transcriptional regulation or $Ca^{2+}$ depletion, because it acts very fast without interfering with other cellular processes. Moreover, LInDA benefits from the precise manipulation with a light as an external trigger, resulting in an immediate disruption of the force bearing complex. Due to the rapid photoinduced cleavage of the CIDs used for LInDA, the kinetics of the dissociation, although not reversible, are significantly faster than it would be possible with optogenetic systems, which are generally based on light-induced gene expression, protein degradation or recruitment of small effector proteins that trigger downstream protein (de-)activation (recently reviewed in Krueger et al.[25]). In contrast, LInDA directly disassembles the key structural components of the AJ complex. However, the distinct activation wavelengths would allow to combine LInDA with optogenetic tools that activate cell contractility[26] or migration[27]. A recently published system for E-cadherin cleavage, based on the photocleavable protein PhoCl[28] is dramatically hampered by the need of high light doses and inefficient dissociation[29].

The central role of cell–cell contacts in orchestrating cell collective behavior has particular evidence during embryonic development and wound healing[30]. AJs mediate the spatial reorganization of cell collectives through a mechanical coupling to cell-matrix adhesion complexes which allows coordinated movement[31]. Here we regulate epithelial collective motility and increase or decrease cellular forces by creating the link between E-cadherin and α-catenin and disrupting it. In comparison to cells which are lacking this molecular link, we observe 48% increase in traction forces (from $1.4 \pm 0.2$ Pa to $2.1 \pm 0.4$ Pa) and 523% increase in cellular tensions (from $8.1 \pm 5.5$ Pa µm to $50.8 \pm 6.4$ Pa µm) when the chemical dimerizer allows the interaction between E-cadherin and α-catenin. A decrease of 25% in traction forces (from $2.1 \pm 0.4$ Pa to $1.7 \pm 0.4$ Pa) and 84% in cellular tensions (from $50.8 \pm 6.4$ Pa µm to $27.5 \pm 6.6$ Pa µm) over time takes place following the targeted photocleavage of the dimerizer and subsequent release of α-catenin in the cytosol.

LInDA allows to control the recruitment of α-catenin and β-catenin at AJs and the reorganization of cytoskeletal fibers which participate in the mechanical stability of cell–cell adhesions. As E-cadherin-mediated adhesions mature, receptor oligomerization and formation of nanoclusters, related to coupling to the actin cytoskeleton, further confer stability to the contacts[32]. Using siRNA knockdown approaches distinct mechanical features have been related to the expression and function of E-cadherin and P-cadherin[9], and forces at cell–cell adhesions might be further regulated by the specific recruitment of AJ proteins. In particular, α-catenin, by acting as switch for the binding of β-catenin and regulation of actin assembly, is crucial for force transmission at AJs[33]. Cytoskeletal rearrangements are crucial for cell–cell adhesion strengthening and only recently the mechanical regulation and generation of interjunctional stress during AJ

biogenesis has been characterized[34]. With LInDA it will be possible to unravel the underlying molecular mechanisms that regulate the early steps of AJ biogenesis in vivo and in vitro while preserving the function of other cell junctional components. In future studies, LInDA may find applications for determining how the different molecular compartments at AJs coordinate each other during contact assembly and disassembly and integrate mechanical forces.

In conclusion, our results show the potential of LInDA to study the cellular response to AJ assembly and disassembly on different scales of space and time, ranging from subcellular precision and sub-second resolution to monolayer compaction and migration and epithelium integrity in vivo over several hours. The possibility of patterned deactivation and combination with specialized imaging techniques like TFM makes LInDA a powerful tool to study the mechanobiology of AJs and its contribution to cellular jamming and unjamming, collective migration and stress propagation in epithelial cell layers both in vitro and in vivo. As such, LInDA may find applications in cell migration and adhesion studies in the context of developmental biology, wound healing and cancer metastasis.

## Methods

**Chemical synthesis**. Detailed information about synthesis and characterization of all dimerizer are given in Supplementary Information. Dimerizers were characterized by NMR and HR-MS/MS.

**Plasmids**. The plasmid for the chimeric E-cadherin/α-catenin construct EcΔ-GFP-α(280-906)[5] was a kind gift from Sergey M. Troyanovsky (Northwestern University, Chicago). E-cadherin-GFP was a gift from Jennifer Stow (Addgene plasmid # 28009; http://n2t.net/addgene:28009; RRID:Addgene_28009)[35]. Monocistronic E-cadherin-Δcyto-Halo (pDO36) was cloned by replacing the cytosolic domain of E-cadherin and GFP in E-cadherin-GFP. The complementary monocistronic DHFR-cyto (pDO38) was made by inserting the cytosolic domain of E-cadherin into DHFR-EGFP. The bicistronic vector coding for E-cadherin-Δcyto-Halo and SNAP-ΔN-α-catenin (pDO56) was created by inserting sequences for Halo with stop codon, IRES and SNAP-mCherry into EcΔ-GFP-α(280-906) via Gibson Assembly[36]. The bicistronic vector coding for E-cadherin-Δcyto-Halo and DHFR-cyto (pDO68) was generated by replacing the GFP-α(280-906) sequences with Halo with stop codon followed by IRES and DHFR-EGFP-cyto in EcΔ-GFP-α(280-906) via Gibson Assembly. Further details are given in Supplementary Information.

**Cell culture and generation of stable cell lines**. Wild type A431 human epidermoid carcinoma cells, MDA-MB 468 human breast cancer cells were purchased from ATCC (A431 CRL-1555; MDA-MB 468 HTB-131), A431 α-catenin KO cells were a kind gift from Takuya Kato (Francis Crick Institute, London) and MDCK E-cadherin KO cells were generously provided by René-Marc Mège and Gautham Hari Narayana (Université Paris Diderot and CNRS, Paris). Cells were maintained in growth medium consisting of Dulbecco's modified Eagle's medium (DMEM), high glucose supplemented with GlutaMAX, sodium pyruvate, 10% fetal bovine serum (FBS) and 1% penicillin–streptomycin, at 37 °C with humidified 5% $CO_2$ atmosphere.

Cells were transfected with Amaxa Cell line Nucleofector (Lonza) following the manufacturer's protocol, selected with Geneticin and sorted by FACS. Kit T (program X-001) was used for A431 α-catenin KO cells stably expressing E-cadherin-Δcyto-Halo and SNAP-ΔN-α-catenin or EcΔ-GFP-α(280-906). For MDA-MB 468 cells kit V (program X-005) was used. Kit L (program L-005) was used to transfect MDCK E-cadherin KO cells with the bicistronic vector coding for E-cadherin-Δcyto-Halo and DHFR-cyto Stable transfected cells were maintained in selection medium (DMEM, high glucose supplemented with GlutaMAX, sodium pyruvate, 10% FBS and 750 µg/ml active Geneticin). Prior to each experiment, cells were FACS-sorted (BD FACSMelody), to ensure cell population homogeneity.

**Dimerization and photocleavage experiments**. Dimerizers were diluted in imaging medium (1:1000 v/v) to a final concentration of 200 nM for Ha-pl-BG, 40 nM for Ha-BG and 200 nM for Ha-pl-TMP, and added to the cells under standard culture conditions.

Photocleaving experiments were performed under live cell imaging conditions (see below). For dimerizer photocleavage in the full field of view, standard DAPI illumination was used (Ex 350/50 nm, 4 × 20 ms, ×63 objective, light source: X-Cite 200DC, 200 W). For photocleavage in defined patterns a 405 nm laser (Leica Infinity Scanner FS, 40 mW and QSP-T filter cube) was used with intensities adjusted for each optical setup: for the ×10 objective, 50% laser power, scan speed 10, spot size 1.54 µm, ND 100%, 20 iterations; for the ×63 objective, 10% laser

power, scan speed 10, spot size 0.68 μm, ND 100%, 20 iterations. Samples for Western blot or immunofluorescence analysis after photocleavage were placed under a UV bench lamp (15 W, 365 nm) with 2 cm distance for 10 min and directly lysed or fixed afterwards.

Each experiment performed in this study was repeated 3–5 times and for imaging experiments 8–10 different image fields were acquired.

**Western blotting.** Cells were lysed in RIPA buffer supplemented with protease/phosphatase inhibitors and 5 mM EDTA (all Thermo Fisher Scientific). SDS-PAGE and Western Blot was done according to the NuPAGE standard protocol using 4–12% Bis-Tris gels and MOPS SDS running buffer. Proteins were blotted on activated nitro cellulose membrane. The membrane was blocked with TBS-T (50 mM TrisHCl, 150 mM NaCl, 0.1% Tween20, pH 7.5) +3% bovine serum albumin (BSA). The membrane was then incubated with the antibodies in 1% bovine serum albumin (BSA) in TBS-T. Primary antibodies were mouse-anti-α-catenin (BD 610193, diluted 1:1000) and mouse-anti-β-actin (Sigma A1978, diluted 1:2000). Secondary antibody was a horseradish peroxidase conjugated anti-mouse antibody produced in goat (Santa Cruz SC-2005, diluted 1:5000). Western blots were developed using ECL Primer Western Blotting Detecting Reagents (GE Healthcare). β-actin served as loading control.

**Immunofluorescence.** Cells were washed with phenol red-free DMEM without supplements, fixed for 20 min with 4% paraformaldehyde at room temperature. Afterwards, cells were washed three times with phosphate buffered saline (PBS), permeabilized with 0.1% Triton X-100 in PBS for 5 min at room temperature, blocked with 1% BSA in PBS for 1 h and incubated with the primary mouse-anti-β-catenin antibody (BD 610153, diluted 1:100) or the primary rabbit-anti-α-catenin antibody (Sigma C2081, diluted 1:200) in 1% BSA in PBS overnight at 4 °C. The next day, cells were washed three times for 10 min with 1% BSA in PBS before adding the secondary goat anti-mouse antibody conjugated with Alexa 350 (Thermo Fisher Scientific A-21049, diluted 1:200) or the secondary goat anti-rabbit antibody conjugated with Alexa 350 (Thermo Fischer Scientific A-21068 diluted 1:200) and phalloidin-Alexa 647 (Thermo Fisher Scientific, final concentration 2 μg/ml). After washing three times for 10 min, samples were mounted in Mowiol. Images were taken with a Leica DM6000B upright fluorescent microscope equipped with a CCD camera (Leica DFC365 FX) and a ×63 oil objective (HC PL APO, NA 1.4). Alternatively, samples were imaged with an Olympus IX inverted microscope and acquisition of images was performed with a DeltaVision system (Applied Precision) using a ×60/1.3 NA plan-Neofluar objective (Olympus) using a cooled CCD camera (Photometrics). Note that E-cadherin and α-catenin were detected by recording the GFP and mCherry fluorescence signal, respectively.

The same procedure was followed to stain for desmosomal marker, but after blocking cells were incubated with primary antibodies mouse-anti-cytokeratin (Millipore MAB3234, diluted 1:100) and guinea pig-anti-desmoplakin (Progen DP-1, diluted 1:100) in 1% BSA in PBS overnight at 4 °C. Secondary antibodies goat anti-mouse conjugated with Alexa 350 (Thermo Fisher Scientific A-21049, diluted 1:200) and donkey anti-guinea pig antibody conjugated with Alexa 647 (Thermo Fisher Scientific A-21235, diluted 1:200) but no phalloidin-Alexa 647 were used. Images were taken with a Zeiss AxioObserver LSM 880 confocal microscope with a photomultiplier tube and a ×63 oil objective (Plan-Apochromat ×63/1.4 Oil DIC M27).

**Live cell imaging.** Cells were seeded in channel slides (μ-Slide VI 0.4, ibidi) or glass bottom culture dish (#1.5 coverslip, ibidi) and maintained in imaging medium (FluoroBrite DMEM, high glucose supplemented with GlutaMAX, sodium pyruvate, 10% FBS and 1% penicillin–streptomycin or 750 μg/ml active Geneticin) at 37 °C with humidified 5% CO$_2$ atmosphere during all live cell imaging experiments. Cells were imaged with a Leica DMi8 inverted fluorescent microscope equipped with a sCMOS camera (Leica DFC9000GT) using either ×10 objective (HC PL FLUOTAR, NA 0.32, PH1), or ×63 objective (HC PL APO CS2, NA 1.40 OIL UV).

**Collective migration analysis.** For measuring the collective monolayer migration, 1 well silicone culture-inserts (ibidi, cut 2 well inlet, growth area 0.22 cm$^2$) were placed in a glass bottom culture dish (#1.5 coverslip, ibidi). $4 \times 10^4$ cells were seeded in 80 μl growth medium and incubated overnight in standard culture conditions. The growth medium in the inlet was replaced with growth medium containing the dimerizer and incubated for 4–6 h. The inlet was removed and cells were carefully washed with phenol red-free DMEM and the dish well was finally filled with imaging medium with or without the dimerizer.

Images were taken with the 10x objective 2 × 2 binning in 5 min intervals starting 1 h after removing the inlet. To analyze the collectivity of monolayer cell migration, the velocity correlation length was used. The displacement vectors between two consecutive images were calculate using a custom-made PIV algorithm[37]. Images were subdivided into 32 × 32 pixel windows with 50% overlap. The displacement vectors were calculated with 16 pixel nodal distance (equals 20.7 μm grid space) using cubic splines based interpolation. The velocity correlation length was then calculated as described previously[21,38] using a custom

made MatLab script with modifications to analyze time lapse recordings. Dividing the displacement vectors by the time difference between two frames yields the velocity vectors $\mathbf{r}_{i,j}$, which were assigned to the central coordinated $(i, j)$ of each 16 × 16 window. Because the axial migration is highly correlated in the described experimental layout, only the lateral component ($U_{i,j}$, perpendicular to the dominant migration direction) was used to calculate the velocity fluctuations along the migration front $u_{i,j}$ as.

$$u_{i,j} = U_{i,j} - \sum_{i=1,m}\sum_{j=1,n}\frac{U_{i,j}}{m \times n} = U_{i,j} - U_{mean}, \quad (1)$$

Thereby, $U_{mean}$ is the mean velocity along the migration front. The lateral correlation function was calculated as

$$C_r = \frac{\langle \mathbf{u}(r') * \mathbf{u}(r' + r)\rangle_{r'}}{\sqrt{\langle \mathbf{u}(r')^2\rangle * \langle \mathbf{u}(r' + r)^2\rangle}}, \quad (2)$$

with $\langle \dots \rangle$ being the average and $\mathbf{r} = \|\mathbf{r}_{i,j}\|$, which is the norm of $\mathbf{r}_{i,j}$. The point where the lateral correlation function (2) reaches the lower threshold 0.01 was defined as the velocity correlation length.

Velocity correlation length with and without dimerizer (Fig. 5a) or before and after 405 nm exposure (Fig. 5b) were compared via unpaired $t$ test with Welch's correction.

**Traction force microscopy and analysis of tension.** Preparation of gels for traction force microscopy (TFM) was done as described previously[39,40]. In brief, polyacrylamide (PAA) gels (Young's modulus 200 Pa) containing 0.5 μm dark red fluorescent carboxylated polystyrene beads (Fluoresbrite 641, Polysciences) were casted on metacrylate activated glass bottom dishes (#0 coverslips, Cellvis). The gel surfaces were coated with 100 μg/ml collagen type I using the photo-reactive cross-linker Sulfo-SANPAH to facilitate cell adhesion. Silicone culture inserts were coated with BSA and carefully placed on the collagen-coated PAA gels. As described for collective migration analysis, $4 \times 10^4$ cells were seeded in 80 μl growth medium and incubated in standard culture conditions until cells attached to the surface. The dimerizers were added and incubated for 4–6 h or overnight to allow the cells to form an intact monolayer. One hour prior to imaging, the inlet was removed and the cells were washed with phenol red free medium.

The cells and beads were imaged for 2–4 h and photocleavage of the dimerizer using the 405 nm laser was done as described above. Afterwards, the cells were trypsinized and bead positions in the relaxed gel were recorded as reference images. Traction force analysis was done following the procedure reported in[38,41] using a 46 × 46 grid space (i.e., 20.72 μm). Tensions were derived from TFM data according to refs. [22,42]. The mechanical stress tensor $\sigma$ was estimated by Bayesian inversion stress microscopy (BISM, see ref. [22] for additional details) from the traction force vector field $\tilde{\mathbf{t}}$ obtained by TFM, using a dimensionless regularization parameter $\Lambda = 10^{-6}$. Given the directionality of monolayer expansion, we imposed free stress boundary conditions on the cell-free boundary only (right hand side boundary on Fig. 6). Specific boundary conditions were not imposed on the three other boundaries: as shown in ref. [42], this does not affect the reliability of stress estimates away from the boundaries.

Calling $x$ and $y$ the coordinates perpendicular and parallel to the moving front, respectively, we observed that the cell sheet is approximately invariant under translations along $y$. This allowed to compute by integration the 1D monolayer tension $\tau^{1D}$:

$$\tau^{1D}(x, t) = \int_{L(t)}^{x}\langle t_x(x', t)_y \mathrm{d}x'\rangle \quad (3)$$

where $L(t)$ was the average position of the free margin at time $t$, and the $y$-average of a field $f$ was defined as

$$\langle f\rangle_y(x, t) = \frac{1}{L_y}\int_0^{L_y} f(x, y, t)\mathrm{d}y, \quad (4)$$

on a domain of extension $L_y$ along $y$. In Supplementary Fig. 9, we compared tension values $\tau^{1D}(x, t)$ obtained by direct integration with estimates $\langle \sigma_{xx}(x, t)\rangle_y$ obtained by averaging over $y$ the $x$ normal component of the stress tensor $\sigma$. Good agreement between the two estimates further validated BISM in the case of an expanding monolayer.

The 2D tension $\tau = \tau^{2D}$, plotted in Fig. 6c, is defined as the isotropic stress:

$$\tau = \frac{\sigma_{xx} + \sigma_{yy}}{2}. \quad (5)$$

The average monolayer tension was computed in bulk, discarding regions of width 200 μm along all boundaries. Instantaneous profiles of the $y$-averaged 2D tension $\langle \tau\rangle_y$ are plotted as a function of the $x$ coordinate in Fig. 6d. Comparison with 1D tension are shown in Supplementary Fig. 9.

**Dimerization and photocleavage experiments in vivo.** DHFR-cyto and E-cadherin-Δcyto-Halo were cloned into pCS2+ vectors. Synthesis of mRNA

transcripts was performed as previously described[43]. Briefly, vectors were linearized with NotI and transcribed with an SP6 RNA polymerase. mRNAs for DHFR-cyto and E-cadherin-Δcyto-Halo were co-injected into both blastomeres of two-cell stage *Xenopus laevis* embryos, as previously described[43]. 4.6 ng of E-cadherin-Δcyto-Halo mRNA and 1.8 ng of DHFR-cyto mRNA were injected per embryo.

Embryos were maintained under standard conditions and staged as previously described[44]. Embryos were incubated in 2 nM Ha-pl-TMP or Ha-TMP from stage 10.5.

Fluorescence imaging of embryos was performed on a Leica SP8 confocal microscope. For photocleavage, a 405 nm laser at 100% power was applied for 30 s followed by 9.5 min intervals over the course of 60 min time.

**Reporting summary**. Further information on research design is available in the Nature Research Reporting Summary linked to this article.

## Data availability

The data supporting the findings of this study are available within the article and Supplementary Information or from the authors upon request. A reporting summary for this article is available as Supplementary Information file.

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

## Acknowledgements

The authors thank H. Rudy for HR-MS/MS analysis; T. Timmermann for NMR measurements; S. M. Troyanovsky for the EcΔ-GFP-α(280–906) plasmid; D. Chenoweth and M. Lampson for the pERB217 plasmid; T. Kato for A431 a-catenin KO cells; R.M. Mege for MDCK E-cadherin KO cells; M. Langlotz and the ZMBH FFCF for cell sorting; B. Koch and C. Sahm for cell sorting and technical support; L. Nechyporenko for help in sample preparation for microscopy; D. Probst and T. Das for providing software for analyzing collective migration; U. Schwarz, D. Probst, J. Di Russo, M. Vishwakarma, and C. Pérez-González for helpful discussion on collective migration and TFM and R. Medda for support and helpful discussion on this work. D.O. and C.Z. are members of the Heidelberg Biosciences International Graduate School (HBIGS). D.O. acknowledges the help of S. Kaspar for software modifications. R.W. and J.P.S. are members of the cluster of excellence CellNetworks at Heidelberg University. D.O., T.P. C.Z., J.P.S. and E.A.C.A. acknowledge support from the Max Planck Society. R.W. acknowledges funding from the

Deutsche Forschungsgemeinschaft DFG (SPP1623, WO 1888/1–2); E.A.C.A. acknowledges funding from BW Stiftung (BiofMO 3D MOSAIC).

## Author contributions

D.O., R.W. and E.A.C.A. conceived the project. D.O, A.S. and C.Z. designed the experiments. D.O. synthesized the dimerizer, cloned the fusion constructs and made stable cell lines. D.O. and T.P. did the IF and WB analysis with help from C.Z. D.O. and T.P. performed all live cell experiments and analyzed the data; A.S. did the in vivo experiments; P.M. analyzed the TFM and tension data for migrating monolayers. R.W. and E.A.C.A. supervised the project. D.O., C.Z., J.P.S., R.M. R.W. and E.A.C.A. made intellectual contributions. D.O., C.Z., A.S., P.M., R.W. and E.A.C.A. prepared the figures. D.O., P.M. R.M. R.W. and E.A.C.A. wrote the paper.

## Competing interests

The authors declare no competing interests.
