## [Peer Review File · Nature Communications]

Reviewers' Comments:

Reviewer #1:

Remarks to the Author:

In this study, Ollech et al. developed a photochemical tool to manipulate adherens junctions. In particular, they managed to control the dissociation of cell-cell junctions and correlate this with physical measurements such as cell velocity and traction forces at various length scales. Overall, they provide an elegant, novel and interesting technique to manipulate cell cohesiveness. I could recommend the publication of the paper since it would be of interest for a large community. However, I found the paper too preliminary for some aspects. I would recommend to clarify and strengthen some of the experiments presented here before publication.

- 1- Statistics are missing in a large part of the experiments. The reproducibility of the results should be clarified. Are the data (for instance, fig1, 2) from a single experiment? Along the line, it will be of interest to discuss and present the variations from different experiments.
- 2- The analysis of line scan profiles (fig.2b) should be improved.
- 3- The results of the correlation are not clearly presented. Is it the velocity correlation length? Moreover, it seems that these measurements gave two very different values for cells treated with Ha-pl-BG in fig 3a and 3b. Why is it so? How reproducible is it?
- 4- The traction force data are disappointed and too preliminary. It is important to strengthen this part. Based on their previous studies, the authors should be able to analyse traction forces and stresses. It will be important to analyze the relationship between the intercellular stress and the position along the migrating monolayers as well as the stress variations over time. Do traction forces on the substrate reinforce over time after UV induction?

Reviewer #2:

Remarks to the Author:

Ollech et al. describe the use of chemical inducers of dimerization (CIDs) to investigate E-cadherin / α -catenin interaction dynamics. Modified E-cadherin and α -catenin (ablation of β -catenin-binding sites in both) were fused to Halo- and SNAP-tag proteins. The authors then claim to have developed a novel assay called LInDA (Light Induced Dissociation of Adherens Junctions).

In this assay, they utilize two dimerizers, the photocleavable small molecule Ha-pl-BG and the photostable Ha-BG. It takes some reading until it becomes clear that the approach is not novel, but that Ha-pl-BG and Ha-BG are simplified and probably less cell permeable versions of optimized CIDs published before (see MeNV-HaXS in {Zimmermann, 2014, 24677313=PMID} and HaXS8 in {Erhart, 2013, 23601644}).

The "marketing" of LInDA suggests a novelty where there is none, Zimmermann et al. {2014, 24677313} already illustrate that the Halo-/SNAP-Tag dimerizers can be used for a wide range of target molecules and cellular localizations. Ha-pl-BG and Ha-BG are not fully characterized CIDs (cell permeability and completeness of dimerization), as only rudimentary control experiments for dimerization are shown.

The biology of the adherence junction formation is interesting, and there are many open questions why/how cell-cell contacts break down when the cadherin/ β -catenin/ α -catenin/ cytoskeletal links are intercepted. The use of CIDs seems promising in this context, as it allows complementary

approaches as compared to knock-down or mutational approaches. The authors also convincingly demonstrate, that this CID approach (as described already in {Zimmermann, 2014, 24677313}) can be used to target the chemically linked complexes in a subcellular fashion.

The combination of traction force microscopy with the CID approach is interesting, and provides a starting point to elucidate the adhesive forces maintaining a cell monolayer. Unfortunately the authors remain at a technical and descriptive level and do not address novel biological concepts, and do not fully exploit the possibilities the CID approach opens to study adherence junction dynamics in space and time.

Reviewer #3:

Remarks to the Author:

Ollech et al., developed a tool that allows the dynamical control of E-Cadherin expression in time and space– LInDA (light induced dissociation of AJs), a new opto-chemical tool that enables a specific “ON-OFF switch” for the mechanical coupling between cells. In their studies, the authors claim that LinDA has the potential to be used in the study of monolayers mechanics and also mention its potential to be coupled to microscopy tools such as TFM. The development of a tools such as LinDA is very timely in several fields. Mostly, because to date there is no tool that allow such versatile manipulation of the AJs complex in time and space. Hence, LinDA has the potential to fill this technical gap in fields studying, cell mechanics, epithelial-to-mesenchymal transition, cell migration, and differentiation, in both, physiology and disease. I believe that this tool could influence the field by providing a tool to address issues that would otherwise be overlooked due to technical limitations.

Although I am very positive about this tool or the potential that this tool could have in the field, I believe that it requires further development in order to meet today’s expectations or requirements of the AJs field. In this context I do have two major points that I believe the authors should address before acceptance:

1. I would like to suggest the authors to invest some time and generate a split E-cadherin LinDA tool. The reason for this is because the current status of the tool pushes the researchers to use or generate a cell line that do not express E-cadherin and/or α -catenin. In some contexts, this may lead to an extreme manipulation of native conditions. The authors could for instance use a form where the cytosolic domains of E-cadherin would replace the α -catenin component of the current tools. This will be very important as this tool could be easily translated to in vivo systems, where the manipulation of endogenous proteins needs to be minimized to keep the system as native as possible. In the same line, this would be relevant when studying EMT in cancer cell lines, where native conditions are also important.
2. Once and if this new version is generated, it would be important to test it in various cell lines and perhaps in a living organism: *Drosophila*, zebrafish, *Xenopus*, etc.

Addressing these issues will deeply expand the impact that this tool could have in several fields and also its ease of use without extra modifications of the native environment. Additionally, the authors claim that this tool could be used to study mechanics of monolayers and they really focus on that subject, but I strongly believe that a powerful tool to modify Cadherin in time and space is necessary in several other biological processes and as it LinDA stands now, the authors will not be exploiting the whole potential that such a tool could have. Thus, I believe that modifications and tests suggested will definitely take LinDA to the level that the field expect.

Responses to reviewer comments

Reviewer #1

Opening remarks- *In this study, Ollech et al. developed a photochemical tool to manipulate adherens junctions. In particular, they managed to control the dissociation of cell-cell junctions and correlate this with physical measurements such as cell velocity and traction forces at various length scales. Overall, they provide an elegant, novel and interesting technique to manipulate cell cohesiveness. I could recommend the publication of the paper since it would be of interest for a large community. However, I found the paper too preliminary for some aspects. I would recommend to clarify and strengthen some of the experiments presented here before publication.*

We thank the reviewer for his positive remarks on the tool and his suggestions to strengthen our work. Please note that the figure numbering has changed due to inclusion of new data.

Comment 1- *Statistics are missing in a large part of the experiments. The reproducibility of the results should be clarified. Are the data (for instance, fig1, 2) from a single experiment? Along the line, it will be of interest to discuss and present the variations from different experiments.*

Response 1- We apologize for missing the part on the statistics and have included in the revised version the information regarding the number of samples/images analyzed in each experiment and the related statistical analysis. Each experiment was repeated 3-5 times and we imaged cells from different fields (8-10 fields per experiment). For the experiments performed to monitor protein localization and the effects of the dimerizer (immunofluorescence, western blotting) we did not observe variations from different experiments. It should be noted that, although our cells were stably transfected to express the constructs for the dimerization, cells were freshly FACS-sorted every time prior to each experiment to avoid variability from experiment to experiment due to the unavoidable presence of non-transfected cells. We noticed however that the monolayer formation and collective migration of A431 cells are highly dependent on cell density and this leads to variations from different experiments (please see response to comment 3 relative to analysis shown in Figure 5) .

Comment 2- *The analysis of line scan profiles (fig.2b) should be improved.*

Response 2- We improved the analysis of the line scan profile now plotted in Figure 4b and in supplementary Figure 3 for the non-photocleavable control (cells incubated with the Ha-BG dimerizer). We analyzed the normalized alpha-catenin and E-cadherin fluorescence intensities over time along a cross-section of cell-cell contact (This is defined by a line drawn between two adjacent cell shown in Figure 4a). The dynamics of AJs dissociation are shown in Figure 3a and b and we included kymographs for comparison of targeted and untargeted AJs.

Comment 3- *The results of the correlation are not clearly presented. Is it the velocity correlation length? Moreover, it seems that these measurements gave two very different values for cells treated with Ha-pl-BG in fig 3a and 3b. Why is it so? How reproducible is it?*

Response 3- We revised the description on PIV on page 9. As the reviewer indicates, we indeed analyzed the velocity correlation length (as shown in supplementary Figure 5 and based on the analysis performed by Das T et al., Nat Cell Biol 2015. The variability in

correlation length observed in former Figure 3a and 3b (now Figure 5) is due to the variation in initial seeding density. It should be noted that the effect of photocleavage of Ha-pl-BG leads to reduction in correlation length which is comparable to the negative control (orange groups “neg. control” and “405 nm laser”) and the decrease is statistically significant when compared to the dimerized groups. We have included a sentence commenting on this variability.

Comment 4- *The traction force data are disappointed and too preliminary. It is important to strengthen this part. Based on their previous studies, the authors should be able to analyse traction forces and stresses. It will be important to analyze the relationship between the intercellular stress and the position along the migrating monolayers as well as the stress variations over time. Do traction forces on the substrate reinforce over time after UV induction?*

Response 4- We thank the reviewer for encouraging us to improve the part on the analysis of our traction force microscopy experiments. We benefitted from a recently established collaboration with Dr. Philippe Marcq and have now performed the analysis of cellular tractions and intracellular stresses by applying Bayesian inversion stress microscopy (as initially developed in Nier et al, Biophys J 2016). This is shown in main Figure 6, supplementary figures 6, 7 and 9, and described in the main text pages 9-11. Further explanations on the analysis can be also found in the material and methods section and in the supplementary file. Moreover, we could now analyze the traction forces on the substrate over time, and after photocleavage of the dimerizer they decrease only gradually (please see Supplementary Figure 7). For 2D migrating layers, the tension in the monolayer drops from approx. 50 to 27.5 Pa μm after photocleavage

Reviewer #2

Opening remarks- *Ollech et al. describe the use of chemical inducers of dimerization (CIDs) to investigate E-cadherin / α -catenin interaction dynamics. Modified E-cadherin and α -catenin (ablation of β -catenin-binding sites in both) where fused to Halo- and SNAP-tag proteins. The authors then claim to have developed a novel assay called LInDA (Light Induced Dissociation of Adherens Junctions).*

Comment 1- *In this assay, the utilize two dimerizers, the photocleavable small molecule Ha-pl-BG and the photostable Ha-BG. It takes some reading until it becomes clear that the approach is not novel, but that Ha-pl-BG and Ha-BG are simplified and probably less cell permeable versions of optimized CIDs published before (see MeNV-HaXS in {Zimmermann, 2014, 24677313=PMID} and HaXS8 in {Erhart, 2013, 23601644}).*

Response 1- Our overall goal was to develop a tool to manipulate cell-cell contacts with light. Such a tool is of high value for various areas of cell biological research because current methods lack efficiency and have limited degree of control. For this we identified E-cadherin and its intracellular interactions as an attractive target which we hoped to manipulate with chemical dimerizers (CIDs). We are fully aware that there are very similar CIDs that have been published by the Wymann group, and we do cite this works, as we believe, appropriately in the manuscript. Currently there are a number of different CIDs that have been demonstrated to work well for colocalization or dimerization of proteins. The general challenge, however, is to combine CIDs with cellular protein interactions in a way to obtain a

phenotypic difference that can tell something about the cell biology to study. This is particularly difficult for cell-cell contacts, which are based on an enormously high degree of complexity of molecular interactions.

Comment 2- *The "marketing" of LInDA suggest a novelty where there is none, Zimmermann et al. (2014, 24677313) already illustrate that the Halo-/SNAP-Tag dimerizers can be used for a wide range of target molecules and cellular localizations.*

Response 2- Obviously, the translation of CIDs into observable changes in cellular behavior (in our case the dissociation of AJs and the loss of cell-cell contacts) is crucial for their usefulness in cell biology. Therefore, LInDA as a tool is not just the application of CIDs; LInDA as a whole is the combination of photocleavable CIDs with appropriately designed E-cadherin and catenin constructs that result in phenotypic differences and can be used for control of cell-cell contacts at an until now not achievable level.

Changes to the manuscript:

In order to address the criticism of Reviewer 2, we have changed the manuscript as follows:

1. We already did reference the photocleavable CID from the Wymann in our initial manuscript. We now also included the reference of the first work about the noncleavable CID from the Wymann lab (Erhart et al. Chemistry and Biology 2013), that had been mentioned by reviewer #2.
2. We further show that the type of CID is less important, as we demonstrate that LInDA works also with Halo-pl-TMP dimerizer. For that we replaced the previously used SNAP-tag by bacterial DHFR which strongly binds to the antibiotic trimethoprim (TMP). LInDA worked with Halo-TMP dimerizers as well as with Halo-BG dimerizers. Being able to use different CIDs in LInDA underlines the importance of the molecular design on the part of the protein for its functionality. In addition, we used Halo-pl-TMP to recruit the cytosolic binding domain of E-cadherin. We were thus able to demonstrate the functionality of LInDA as a tool for the manipulation of AJs *in vivo* (live *Xenopus* embryos). We think the new data shows even more clearly that LInDA is a tool that is, as reviewer #2 mentions, based on existing types of CIDs, but as a tool to manipulate cell-cell contacts, it is a completely new and original tool.
3. We added the information about synthesis and characterization of the additional CIDs Halo-pl-TMP and Halo-TMP to the SI. The CID structures and the molecular design of LInDA is further added to Figure 1 (design and characterization of the novel protein constructs is also added to SI, Supplementary Figure 1).

Comment 3- *Ha-pl-BG and Ha-BG are not fully characterized CIDs (cell permeability and completeness of dimerization), as only rudimentary control experiments for dimerization are shown.*

Response 3- Concerning cell permeability, we would like to point that we use rather low working concentrations in the nanomolar range. The CIDs seem to pass cell membranes quite well. In particular, we can prove the cell permeability by the CID's functionality to result in a strong phenotypic change (further proven by WB, live cell fluorescence microscopy colocalization analysis and immunostaining). The cell-permeability of the CIDs is that good that we can apply LInDA for *in vivo* experiments in *Xenopus* embryos, where cell penetration is in general much more challenging than in cell culture.

Comment 4- *The biology of the adherence junction formation is interesting, and there are many open questions why/how cell-cell contacts break down when the cadherin/b-catenin/a-catenin/ cytoskeletal links are intercepted. The use of CIDs seems promising in this context,*

as it allows complementary approaches as compared to knock-down or mutational approaches. The authors also convincingly demonstrate, that this CID approach (as described already in {Zimmermann, 2014, 24677313}) can be used to target the chemically linked complexes in a subcellular fashion. The combination of traction force microscopy with the CID approach is interesting, and provides a starting point to elucidate the adhesive forces maintaining a cell monolayer. Unfortunately the authors remain at a technical and descriptive level and do not address novel biological concepts, and do not fully exploit the possibilities the CID approach opens to study adherence junction dynamics in space and time.

Response 4- We are glad that the reviewers finds the use of our tool promising for studying the biogenesis of adherens junctions. We addressed his suggestion to go beyond a technical and descriptive level: in the revised manuscript we added new data presented in Figure 4, showing that during the dissociation process, desmosomes are functional in maintaining cell-cell contacts. We also characterized further, as also suggested by reviewer #1, the implications on epithelium mechanics caused by photocleavage and light induced dissociation of AJs (Figure 6). Finally, we applied our tool *in vivo* (Figure 7). After having shown in the current work that our tool allows the manipulation of AJs without altering their function is certainly opening up new applications to address open questions in the field of cell-cell adhesion.

Reviewer #3

Opening remarks- *Ollech et al., developed a tool that allows the dynamical control of E-Cadherin expression in time and space— LinDA (light induced dissociation of AJs), a new opto-chemical tool that enables a specific “ON-OFF switch” for the mechanical coupling between cells. In their studies, the authors claim that LinDA has the potential to be used in the study of monolayers mechanics and also mention its potential to be coupled to microscopy tools such as TFM. The development of a tools such as LinDA is very timely in several fields. Mostly, because to date there is no tool that allow such versatile manipulation of the AJs complex in time and space. Hence, LinDA has the potential to fill this technical gap in fields studying, cell mechanics, epithelial-to-mesenchymal transition, cell migration, and differentiation, in both, physiology and disease. I believe that this tool could influence the field by providing a tool to address issues that would otherwise be overlooked due to technical limitations. Although I am very positive about this tool or the potential that this tool could have in the field, I believe that it requires further development in order to meet today’s expectations or requirements of the AJs field.*

We are glad that the reviewer finds our tool interesting and timely to study epithelial dynamics and function and appreciate her/his encouragement to meet the expectations in the field of AJs.

In this context I do have two major points that I believe the authors should address before acceptance:

Comment 1- *I would like to suggest the authors to invest some time and generate a split E-cadherin LinDA tool. The reason for this is because the current status of the tool pushes the researchers to use or generate a cell line that do not express E-cadherin and/or a-catenin. In some contexts, this may lead to an extreme manipulation of native conditions. The authors could for instance use a form where the cytosolic domains of E-cadherin would replace the a-catenin component of the current tools. This will be very important as this tool could be easily*

translated to in vivo systems, where the manipulation of endogenous proteins needs to be minimized to keep the system as native as possible. In the same line, this would be relevant when studying EMT in cancer cell lines, where native conditions are also important.

Response 1- To test LinDA application in cells while minimizing the manipulation of native conditions, we applied of the E-cadherin/ α -catenin tool in non-knockout cells (MDA-MB-468) as shown in Supplementary Figure 4.

We also followed the suggestion of the reviewer and have generated a split E-cadherin tool. Its design is shown in Figure 1e and g and the Ha-pl-TMP dimerizer used for reconstituting the split cadherin receptor is shown in Figure 1f. The application of this tool is shown in main Figure 7 and supplementary Figure 8. We applied the split cadherin setup *in vivo* in *Xenopus* embryos at developmental stage 10.5 without perturbation of the native conditions. As indicated on page 11, at this stage endogenous E-cadherin starts to be expressed, and the injected E-cadherin- Δ cyto-Halo and DHFR-cyto constructs act as dominant negative, causing epithelial dissociation. The use of the dimerizer rescues this phenotype, indicating that the dimerizer-induced junctions are functional.

Comment 2- *Once and if this new version is generated, it would be important to test it in various cell lines and perhaps in a living organism: Drosophila, zebrafish, Xenopus, etc.*

Response 2- We felt highly motivated in trying our tool *in vivo* and we teamed up with the lab of Roberto Mayor to apply the split E-cadherin construct in *Xenopus* embryos. Please refer to Figure 7 and to response to comment 1.

Comment 3- *Addressing these issues will deeply expand the impact that this tool could have in several fields and also its ease of use without extra modifications of the native environment. Additionally, the authors claim that this tool could be used to study mechanics of monolayers and they really focus on that subject, but I strongly believe that a powerful tool to modify Cadherin in time and space is necessary in several other biological processes and as it LinDA stands now, the authors will not be exploiting the whole potential that such a tool could have. Thus, I believe that modifications and tests suggested will definitely take LinDA to the level that the field expect.*

Response 3- We thank the reviewer once again for his suggestions and show now that LinDA can indeed modify AJs in time and at multiple length scale (please see main Figures 3, 4 and 7). Certainly in future studies it will be interesting to focus on AJ protein dynamics and functions during assembly and disassembly *in vitro* and *in vivo* for a variety of biological processes, and we will be very happy to make LinDA available for the community working in the field of cell-cell and cell-matrix adhesion.

Reviewers' Comments:

Reviewer #1:

Remarks to the Author:

I found authors answers to my comments and questions and to the ones for the two other reviewers clear and convincing. The additional experiments (Traction forces and in vivo) are appealing. Finally, the manuscript is now much clearer, understandable and provides new insights into the efficiency and versatility of LINDA. I support paper acceptance.

I have a few remaining comments:

1/ I think that Supplemental fig 4 should be in the main text.

2/ I would remove this sentence on page6 : "To our knowledge, inducing the dissociation of AJs with such high precision has not been possible so far." (or rephrase it)

3/ I was wondering how the authors compute the correlation length in fig 5b. It seems that the time-scale is short and cells do not exhibit large movements over 2 hours (Suppl Movie 7). The authors could maybe clarify this point. (by the way, this is the velocity correlation length. It should be mentioned).

4/ Fig3c could be better explained in the text.

Reviewer #2:

Remarks to the Author:

The authors have improved the manuscript and added additional results using photocleavable CIDs.

I would still suggest that title and manuscript would not market "LInDA" but refer to photocleavable CIDs and - most importantly - focus on the results that were produced using these molecules.

The clarity of the manuscript, and figure quality are now high.

There are still some points that should be improved: in figure legends it would be preferable when (light)-"pulse" would be replaced by the exact duration and conditions of illumination (light source, power).

The western blot of Figure 2c should be quantified (bands at 265 kD and 199 kD). It is likely to become apparent that only <50% of SNAP-dN-a-cat is dimerized. The authors should document this and explain why a 100% cross-linking is not required here to generate cellular phenotypes.

In videos (for example supp video 1) it should be indicated when dimerizer is added (as a text overlay ?).

In Figure 6, the statement "positions analyzed" is vague. Field of views, or partial field of views shown? Are phase contrast and fluorescence images at equal magnification?

Reviewer #4:

Remarks to the Author:

The authors have fully addressed the points raised by Reviewer #3. There is an erroneous „and“ at the end of the legend of Fig. S14 that the authors might wish to remove.

Responses to reviewers' comments

Reviewer #1

Opening remarks- *I found authors answers to my comments and questions and to the ones for the two other reviewers clear and convincing. The additional experiments (Traction forces and in vivo) are appealing. Finally, the manuscript is now much clearer, understandable and provides new insights into the efficiency and versatility of LINDA. I support paper acceptance. We are glad that the reviewer finds the revised version of our manuscript clear and convincing.*

I have a few remaining comments:

Comment 1- *I think that Supplemental fig 4 should be in the main text.*

Response 1- We moved Supplemental fig 4 to the main text, as part of Figure 2.

Comment 2- *I would remove this sentence on page6 : "To our knowledge, inducing the dissociation of AJs with such high precision has not been possible so far." (or rephrase it).*

Response 2- We removed the sentence.

Comment 3- *I was wondering how the authors compute the correlation length in fig 5b. It seems that the time-scale is short and cells do not exhibit large movements over 2 hours (Suppl Movie 7). The authors could maybe clarify this point. (by the way, this is the velocity correlation length. It should be mentionned).*

Response 3- We agree with the reviewer that it is very difficult to observe the movements over 2 hours, this is unfortunately due to the fact that A431 cells are not highly motile, like e.g. MDCK. Nevertheless, our analysis tool could still record the movement and velocities and the difference between our experimental group and the control group.

We have emended the text and indicated as "velocity correlation length" as suggested by the reviewer.

Comment 4- *Fig3c could be better explained in the text.*

Response 4- We improved the explanation of Figure 3c on page 6.

Reviewer #2

Opening remarks- *The authors have improved the manuscript and added additional results using photocleavable CIDs. I would still suggest that title and manuscript would not market "LInDA" but refer to photocleavable CIDs and - most importantly - focus on the results that were produced using these molecules. The clarity of the manuscript, and figure quality are now high.*

We are glad that the reviewer finds the revised version of our manuscript clear and convincing.

There are still some points that should be improved:

Comment 1- *in figure legends it would be preferable when (light)-"pulse" would be replaced by the exact duration and conditions of illumination (light source, power).*

Response 1- We have inserted the detailed information about the duration and conditions of illumination.

Comment 2- *The western blot of Figure 2c should be quantified (bands at 265 kD and 199 kD). It is likely to become apparent that only <50% of SNAP-dN-cad is dimerized. The authors should document this and explain why a 100% cross-linking is not required here to generate cellular phenotypes.*

Response 2- We apologize with the reviewer for the confusion on the western blot. The quantification is shown in Supplementary Figure 2; we also indicated now in a consistent way in figure 2c and in Supplementary Figure 2 the lane labeling, since these refer to the same blot. The reviewer is right regarding the lower amount of proteins that dimerize: however, as we observed the expected phenotype upon addition of the CID, full dimerization to E-cadherin is not needed to restore adherens junction formation. This is not surprising, since dynamics processes that required receptor complex formation in cell adhesion are based on equilibrium of molecular interactions.

Comment 3- *In videos (for example supp video 1) it should be indicated when dimerizer is added (as a text overlay ?).*

Response 3- We inserted the information in the Supplementary Movie files.

Comment 4- *In Figure 6, the statement "positions analyzed" is vague. Field of views, or partial field of views shown? Are phase contrast and fluorescence images at equal magnification?*

Response 4- We rephrased how we indicate the microscopy images we analyzed, as in fact, randomly selected fields of view are shown. The phase contrast images and the resulting heat maps (these are not fluorescent images) are at equal magnification.

Reviewer #4

The authors have fully addressed the points raised by Reviewer #3. There is an erroneous „and“ at the end of the legend of Fig. S14 that the authors might wish to remove.

We thank the reviewer for the positive remark and we removed the erroneous word at the end of the legend.